# Using Principal Component Analysis and RNA-Seq to Identify Candidate Genes That Control Salt Tolerance in Garlic (*Allium sativum* L.)

Xiangjun Zhou [1], Yanxia Dou [1], Xiaoxia Huang [1], Gang Li [1], Hongrui Zhang [1], Dagang Jiang [1], Jinping Fan [1], Jorge Alberto Condori-Apfata [1], Xiaoqin Liu [1], Sandro Jhonatan Condori-Pacsi [2], Roxana M. Bardales [3], Mayela Elizabeth Mayta Anco [3], Helbert O. Lazo [2], Herbert Angel Delgado Salazar [2], Maria Valderrama Valencia [2] and Cankui Zhang [1,*]

1. Department of Agronomy and Center for Plant Biology, Purdue University, West Lafayette, IN 47907, USA; zhou611@purdue.edu (X.Z.); douyanxia@swu.edu.cn (Y.D.); huangxx@swfu.edu.cn (X.H.); gangli@sdau.edu.cn (G.L.); zhanghongrui2003@126.com (H.Z.); dagangj@scau.edu.cn (D.J.); jinpingfan@neau.edu.cn (J.F.); jacondor@purdue.edu (J.A.C.-A.); liu2617@purdue.edu (X.L.)
2. Departamento Académico de Biología, Universidad Nacional de San Agustín, Arequipa 04001, Peru; scondorip@unsa.edu.pe (S.J.C.-P.); hlazor@unsa.edu.pe (H.O.L.); hdelgados@unsa.edu.pe (H.A.D.S.); mvalderramav@unsa.edu.pe (M.V.V.)
3. Facultad de Agronomía, Universidad Nacional de San Agustín, Arequipa 04001, Peru; rbardales@unsa.edu.pe (R.M.B.); mmaytaan@unsa.edu.pe (M.E.M.A.)
* Correspondence: ckzhang@purdue.edu

**Abstract:** To examine physiological responses of garlic to salinity, 17-day-old seedlings of eight soft-neck accessions were treated with 200 mM NaCl for seven days in a hydroponic system. Several morphological and physiological traits were measured at the end of the treatment, including shoot height, shoot fresh weight, shoot dry weight, root length, root fresh weight, root dry weight, photosynthesis rate, and concentrations of $Na^+$ and $K^+$ in leaves. The principal component analysis showed that shoot dry weight and $K^+/Na^+$ ratio contribute the most to salt tolerance among the garlic accessions. As a result, salt-tolerant and sensitive accessions were grouped based on these two parameters. Furthermore, to investigate the molecular mechanisms in garlic in response to salinity, the transcriptomes of leaves and roots between salt-tolerant and salt-sensitive garlic accessions were compared. Approximately 1.5 billion read pairs were obtained from 24 libraries generated from the leaves and roots of the salt-tolerant and salt-sensitive garlic accessions. A total of 47,509 genes were identified by mapping the cleaned reads to the garlic reference genome. Statistical analysis indicated that 1282 and 1068 genes were upregulated solely in the tolerant leaves and roots, whereas 1505 and 1203 genes were downregulated exclusively in the tolerant leaves and roots after NaCl treatment, respectively. Functional categorization of these genes revealed their involvement in a variety of biological processes. Several genes important for carotenoid biosynthesis, auxin signaling, and $K^+$ transport were strongly altered in roots by NaCl treatment and could be candidate genes for garlic salt tolerance improvement.

**Keywords:** garlic; salt stress; principal component analysis; RNA-seq; transcriptome

## 1. Introduction

With soil salinization threatening more than one-third of cultivated lands, salinity stress becomes one of the most adverse environmental factors that significantly affect plants' growth and development [1]. Over-accumulation of sodium ($Na^+$) and chloride ($Cl^-$) in plant cells has detrimental effects on physiological traits and metabolic processes, such as ion imbalance, water stress, oxidative damage, abnormal membrane stability, and reduced photosynthesis, resulting in plant wilting and death [2–8]. On the other hand, plants have evolved complicated mechanisms to cope with salt stress.

The exclusion of $Na^+$ from the shoot is an essential mechanism for plants to maintain regular photosynthesis rate and growth [3]. High-affinity $K^+$ transporter 1 (HKT1) is a plasma membrane importer responsible for sodium exclusion [9]. $Ca^{2+}$ in plant tissues was also found to increase $Na^+$ exclusion in order to improve salinity tolerance [10]. In addition, root system architecture (RSA) significantly changes to avoid salt stress, which is mediated by plant hormones auxin and abscisic acid (ABA) [11–13]. ABA plays a crucial role in salt tolerance in many plant species by regulating stomatal closure, root growth, and stress-responsive gene expression. Genes involved in ABA biosynthesis, phytoene synthase (PSY), nine-cis-epoxy-carotenoid dioxygenase (NCEDs), ABA DEFEICIENTS (ABAs), and aldehyde oxidase 3 (AAO3) are induced by salinity in various tissues [14–16]. Sucrose non-fermenting-1-related protein kinases (SnRK2.2, SnRK2.3, and SnRK2.6), the central components in the ABA signaling pathway, are activated following the perception of ABA and regulate ion transport, reactive oxygen species (ROS) production, and gene expression by phosphorylating different targets [17,18]. Another well-studied signaling cascade involved in salt stress responses is the salt overly sensitive (SOS) pathway. Its main components, SOS1, SOS2, and SOS3, maintain ion homeostasis in plant cells under salt stress [19–21]. Several SOS orthologs have been identified in rice and Brassica species, indicating that the pathway is conserved in plants [22,23]. Despite common characteristics in adaptation and tolerance of plants to salinity, the dominant mechanism by which plants survive salinity stress could be different from species to species.

Garlic (*Allium sativum* L.) is one of the important seasonings and horticulture plants. A previous study showed that soil salinity drastically reduced garlic yield and quality [24]. Unlike other plant species such as Arabidopsis, rice, wheat, barley, and tomato, limited information is available about garlic's physiological responses under salt stress. Understanding physiological responses of garlic to salt stress and the molecular basis of salt tolerance will help improve the vegetables' abiotic tolerance.

As a member of the Allium family, most garlic varieties are diploids (2 n = 2× = 16). The genome size of the haploid garlic is estimated to be 15.9 giga-base pairs (Gb), 118, 32, and 6 times bigger than the genomes of Arabidopsis, rice, and maize, respectively, indicating a complex genome structure and making garlic genome sequencing challenging [25,26]. Recently, approximately 16.24 Gb of garlic (cultivar Ershuizao) genome were sequenced and assembled into eight chromosomes with 57,561 predicted protein-coding genes [27]. Many garlic genes involved in the biosynthesis of sulfur compounds, salt stress tolerance, development of the flower, pollen and clove, and green discoloration by low-temperature were identified by RNA-sequencing (RNA-seq) technology followed by de novo assembly [28–33].

In this study, eight garlic accessions were hydroponically grown and treated with 200 mM NaCl. Several morphological and physiological traits were measured at the end of treatment. Salt tolerance was evaluated, and salt-tolerant and salt-sensitive accessions were grouped using a principal component analysis (PCA). Subsequently, the transcriptomes of salt-treated leaf and root tissues from the salt-tolerant and sensitive garlic accessions were examined by RNA-seq, identifying a total of 5386 differentially expressed genes in the salt-tolerant leaves and roots. These genes were involved in diverse biological processes that mediate salt tolerance in garlic.

## 2. Materials and Methods

### 2.1. Plant Materials and Growth Conditions

Garlic (*A. sativum* L.) accessions were obtained from the U.S. Department of Agriculture National Plant Germplasm (USDA-NAGS). Their accession numbers, origins, and improvement status were listed in Figure 1A. Healthy garlic cloves were grown in Metro Mix soil in a growth chamber under a 14 h light period with light intensity of 800 μmol/m² at 23 °C. Twelve-day-old garlic seedlings were unearthed, washed, and then transferred to Hoagland solution and grown for another five days. The hydroponic solution was changed every three days.

| ID | Name | Accession No. | Origin | Improvement status |
|---|---|---|---|---|
| 2 | WKP-88-6 | PI 543048 | Pakistan | Cultivar |
| 4 | SILVERSKIN | PI 540375 | US | Cultivar |
| 5 | BLANCO DE HUELMA ZAMORA I | PI 615422 | Spain | Cultivar |
| 6 | 870727 | PI 540379 | Chile | Cultivar |
| 7 | PRISTINSKI | PI 383819 | Serbia | Cultivar |
| 8 | Rose Du Var | W6 35682 | US | Cultivar |
| 10 | 5 | W6 18727 | Hungary | Cultivar |
| 11 | W6 671 | W6 671 | Turkey | Uncertain |

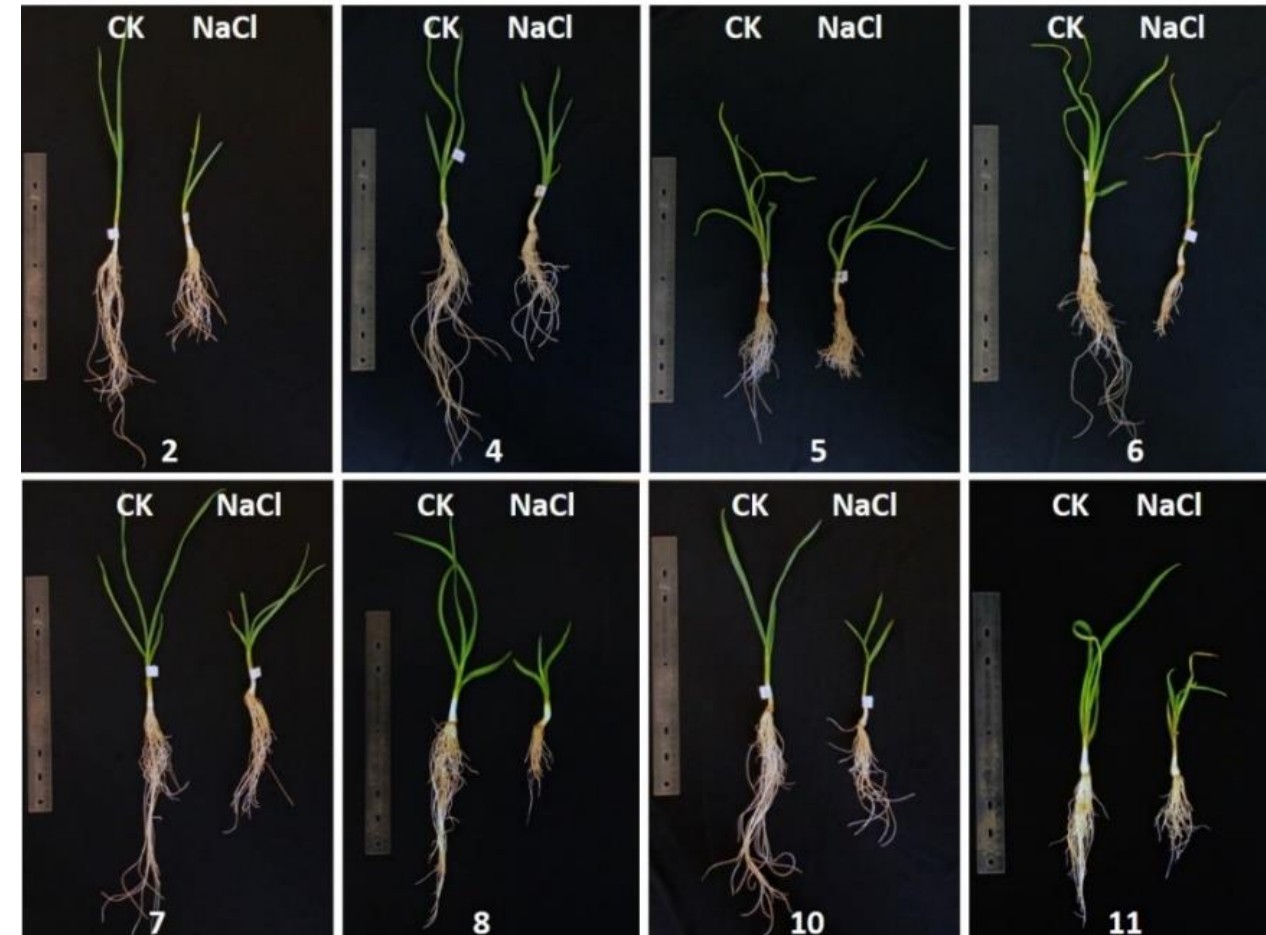

**Figure 1.** Garlic plants under NaCl stress. (**A**). Experimental IDs, names, accession numbers, origins, and improvement status of garlic accessions used in the study. (**B**). Garlic seedlings grown in a hydroponic solution with or without NaCl. The photos were taken seven days after NaCl treatment.

### 2.2. Salt Stress Treatment and Physiological Measurements

For salt treatment, NaCl was added into Hoagland solution to a final concentration of 200 mM while the seedlings grown in Hoagland solution were used as the control. The seedlings were collected seven days later, and shoot height, shoot fresh weight, root length, and root fresh weight were determined. Shoot and root dry weight were measured after the shoot and root were dried at 80 °C for three days. At the end of salt treatment, three control and three salt-treated plants were chosen to measure the photosynthesis rate by using LI-COR 6400XT following the manufacturer's instructions (LI-COR, Lincoln, NE). Some of the seedlings were collected in liquid nitrogen and stored at −80 °C for RNA extraction. For $Na^+$ and $K^+$ measurement, 50 mg of dry powder of garlic leaves were digested in $H_2SO_4$ and diluted to 50 mL by adding ultrapure water. $Na^+$ and $K^+$ concentrations were measured in a plasma atomic emission spectrometer (ICP9820, Shimadzu, Columbia, MD, USA).

### 2.3. Principal Component Analysis (PCA)

The data from the salt treatment were used in the PCA by following the protocol online (https://www.programmersought.com/article/31583857013/, accessed on 7 October 2020). The principal components (PCs) and contributions of each trait to PCs were determined. Based on the two traits that made the biggest contributions to the combined PC1/PC2, eight garlic accessions were grouped.

### 2.4. RNA Extraction, Library Construction and Quantitative Reverse Transcription PCR (qRT-PCR)

RNA was extracted from garlic shoot and root tissues using an E.Z.N.A.® Plant RNA Kit (Omega Bio-Tek Company, Norcross, GA, USA) according to the manufacturer's instructions. RNA samples' integrity was checked in an Agilent 2100 Bioanalyzer System (Santa Clara, CA) to ensure RNA integrity number (RIN) above 8.0. Four mg of total RNA were used to construct RNA-seq libraries with the TruSeq RNA Library Prep Kit (Illumina Company, San Diego, CA, USA). The libraries were sequenced on an Illumina NovaSeq 6000 sequencing platform with the $2 \times 150$ bp paired-end mode. Approximately 1 μg of RNA samples was converted into cDNAs using the iScript™ cDNA synthesis kit (Bio-Rad, Hercules, CA, USA). Expression of the selected genes was examined by qRT-PCR by using the gene specific primers (Supplementary Table S1). Garlic elongation factor 1 alpha (*EF-1α*) was used as internal control. The melt curve analysis was run afterwards to confirm the specificity of PCR amplification. qRT-PCR data indicated the mean of three biological replicates.

### 2.5. RNA-Seq Data Analysis and Functional Annotation

Raw RNA-seq reads were first processed by trimming the adaptor and low-quality sequences using Trimmomatic [34] with default parameters and filtering reads that were mapped to the rRNA database [35] using bowtie [36], allowing up to three mismatches. The cleaned RNA-seq reads were mapped to the garlic reference genome [27] using STAR [37]. Following alignments, raw counts for each gene were calculated and then normalized to fragments per kilobase of exon model per million mapped fragments (FPKM). Differentially expressed genes were identified using DESeq2 [38], with the criteria of log2 (fold change) $\geq 1$ or $\leq -1$, and adjusted *p*-value $\leq 0.05$. GO term annotation of garlic genes and GO term enrichment analysis of differentially expressed genes (DEGs) were performed using the Blast2GO suite [39]. Enriched GO terms were summarized and visualized using REVIGO (http://revigo.irb.hr/, accessed on 10 December 2020) [40]. Pathway annotation was performed using BlastKOALA [41] and significantly enriched pathways in DEGs were identified using the hypergeometric test.

## 3. Results

### 3.1. Physiological Response of Garlic Cultivars to Salt Stress

Seventeen-day-old garlic seedlings from eight soft-neck garlic accessions were treated with 200 mM NaCl for seven days in a hydroponic system (Figure 1). At the end of the treatment, nine traits were measured in garlic seedlings under non-salinity (Hoagland solution) and salinity (Hoagland solution with 200 mM NaCl), including shoot height (SH), shoot fresh weight (SFW), shoot dry weight (SDW), root length (RL), root fresh weight (RFW), root dry weight (RDW), photosynthesis rate, as well as $Na^+$ and $K^+$ concentrations in leaves (Table 1). Salt stress significantly affected garlic growth and development. An average of around 29.7%, 46.9%, 29.8%, 42.4%, 40.7%, 36%, and 77.7% reduction was found for SH, SFW, SDW, RL, RFW, RDW, and photosynthesis rate, respectively. $Na^+$ concentration is relatively low in plant leaves grown under normal conditions and significantly increased under salinity stress [42,43]. The exclusion of $Na^+$ from the shoot is a tolerance mechanism for plants to survive [3]. Therefore, $Na^+$ and $K^+$ concentrations were measured in salt-treated leaves in this study. $Na^+$ and $K^+$ concentrations among these garlic accessions ranged from 14.00–28.19 mg/g DW and 29.34–46.05 mg/g DW, respectively (Table 1), indicating differences among the garlic accessions in the uptake and exclusion of $Na^+$ and $K^+$.

**Table 1.** Mean values of shoot height (SH), shoot fresh weight (SFW), shoot dry weight (SDW), root length (RL), root fresh weight (RFW), root dry weight (RDW), photosynthesis rate (Photo), $Na^+$, and $K^+$ concentrations under normal and salinity conditions.

| Traits | Control | | NaCl 200 mM | |
|---|---|---|---|---|
| | Range | Mean | Range | Mean |
| SH (cm) | 16.5–41.1 | 28.3 | 13.2–29.8 | 19.9 |
| SFW (g/plant) | 1.88–8.59 | 4.07 | 0.61–5.4 | 2.16 |
| SDW (g/plant) | 0.30–0.76 | 0.47 | 0.15–0.58 | 0.33 |
| RL (cm) | 9.6–33.1 | 26.2 | 9.2–21.1 | 15.1 |
| RFW (g/plant) | 1.13–5.21 | 3.0 | 0.4–3.68 | 1.78 |
| RDW (g/plant) | 0.11–0.37 | 0.25 | 0.04–0.27 | 0.16 |
| Photo ($\mu mol\ m^{-2}\ s^{-1}$) | 4.42–18.03 | 11.87 | 1.20–4.68 | 2.65 |
| $Na^+$(mg/g) | ND | ND | 14.0–28.2 | 20.2 |
| $K^+$(mg/g) | ND | ND | 29.3–46.0 | 37.8 |

ND, not detected.

### 3.2. PCA and Separation of Garlic Accessions

PCA is a robust mathematical algorithm that reduces the measurement data's complexity. By comparing a few principal components, samples can be separated and grouped [44]. PCA of the above measurement data showed that the first (PC1), second (PC2), and third principal components (PC3) explained 51.36%, 29.58%, and 8.29% of the variance, respectively. As high as 80.93% of the traits' variance could be explained by combining the PC1/PC2. In the combined PC1 and PC2, shoot dry weight (SDW) and $K^+/Na^+$ ratio displayed the highest loading values. Therefore, they were used to access the salinity tolerance of garlic accessions (Figure 2A). Previous studies showed that salt-tolerant plants often have the higher $K^+/Na^+$ ratio and less biomass reduction [45,46]. As a result, garlic accessions 4 (PI 540375), 7 (PI 383819), 8 (W6 35682), 10 (W6 18727), and 11 (W6 671) were grouped as the tolerant accessions, while 2 (PI 543048), 5 (PI 615422), and 6 (PI 540379) were considered the salinity sensitive accessions (Figure 2B).

**A**

| Traits | PC1 | PC2 | PC1/PC2 |
|---|---|---|---|
| Shoot height | 14.222848 | 0.01779996 | 9.0 |
| Shoot fresh weight | 8.626765 | 16.46258 | 11.5 |
| Shoot dry weight | 18.046 | 1.92154795 | 12.1 |
| Root length | 17.125834 | 0.64047141 | 11.4 |
| Root fresh weight | 16.227437 | 1.37375054 | 10.7 |
| Root dry weight | 13.09124 | 6.71528695 | 10.7 |
| Photosynthesis | 4.438425 | 20.0723586 | 10.2 |
| $Na^+$ | 2.236598 | 26.4967806 | 11.2 |
| $K^+/Na^+$ | 5.384707 | 26.29945210 | 13.1 |
| Variance percentage | 51.35698 | 29.57546 | 80.9 |

**B**

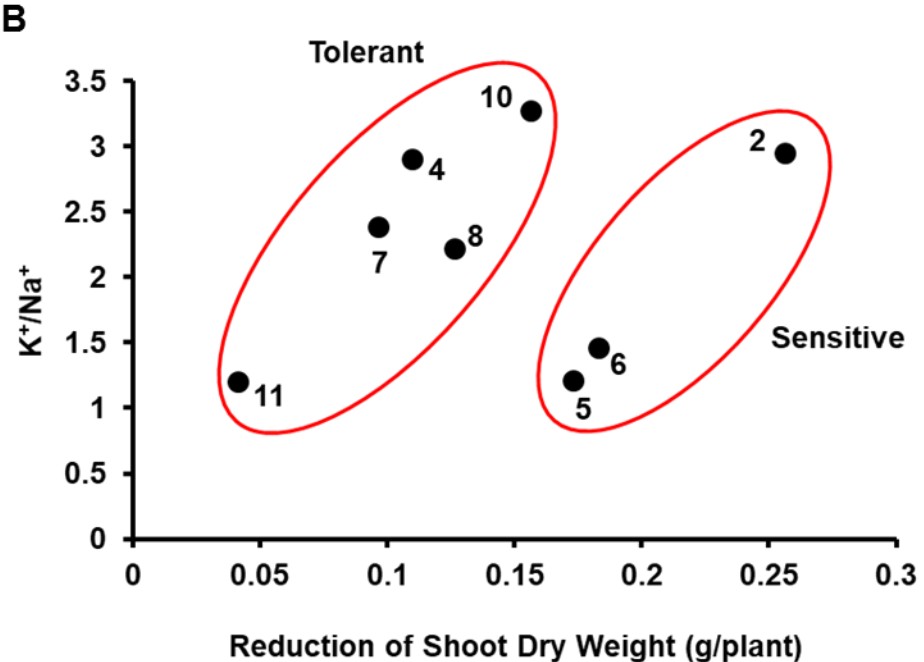

**Figure 2.** Principal component analysis of the physiological traits. (**A**). Contribution (%) of traits to the principal components. (**B**). Separation of garlic accessions according to the shoot dry weight and $K^+/Na^+$.

### 3.3. Comparative Transcriptome in the Salinity-Tolerant and Sensitive Accessions by RNA-Seq

A salinity-tolerant accession (4) and sensitive accession (5) were chosen to study differentially expressed genes in leaves and roots after NaCl treatment. Twenty-four libraries were made from three replicates of the garlic accessions' leaves and roots under normal conditions or salt stress (Supplementary Table S2). A total of 1503,823,225 raw and 1434,595,989 cleaned read pairs were obtained from these 24 libraries. An average of 33.25 million read pairs from each sample were mapped to the garlic assembled genome [27] (Supplemental Table S2), resulting in the identification of 47,509 genes from the RNA-seq analysis.

Statistical analysis with an adjusted p-value below 0.05 showed that 719 and 155 genes were expressed two-fold higher, whereas 726 and 259 genes were downregulated

(two-fold reduction) in the sensitive shoot and root tissues treated with salt, respectively (Figure 3A,B, Supplemental Table S3). In comparison, 1571 and 1391 were upregulated in salt-treated leaves and roots of the salt-tolerant accession, whereas 2045 and 1660 genes were downregulated (Figure 3C,D, Supplemental Table S3). A larger number of differentially expressed genes (DEGs) in the salt-tolerant line than in the sensitive line implied that a broader range of responses occurred in the salt-tolerant garlic accession, especially in roots.

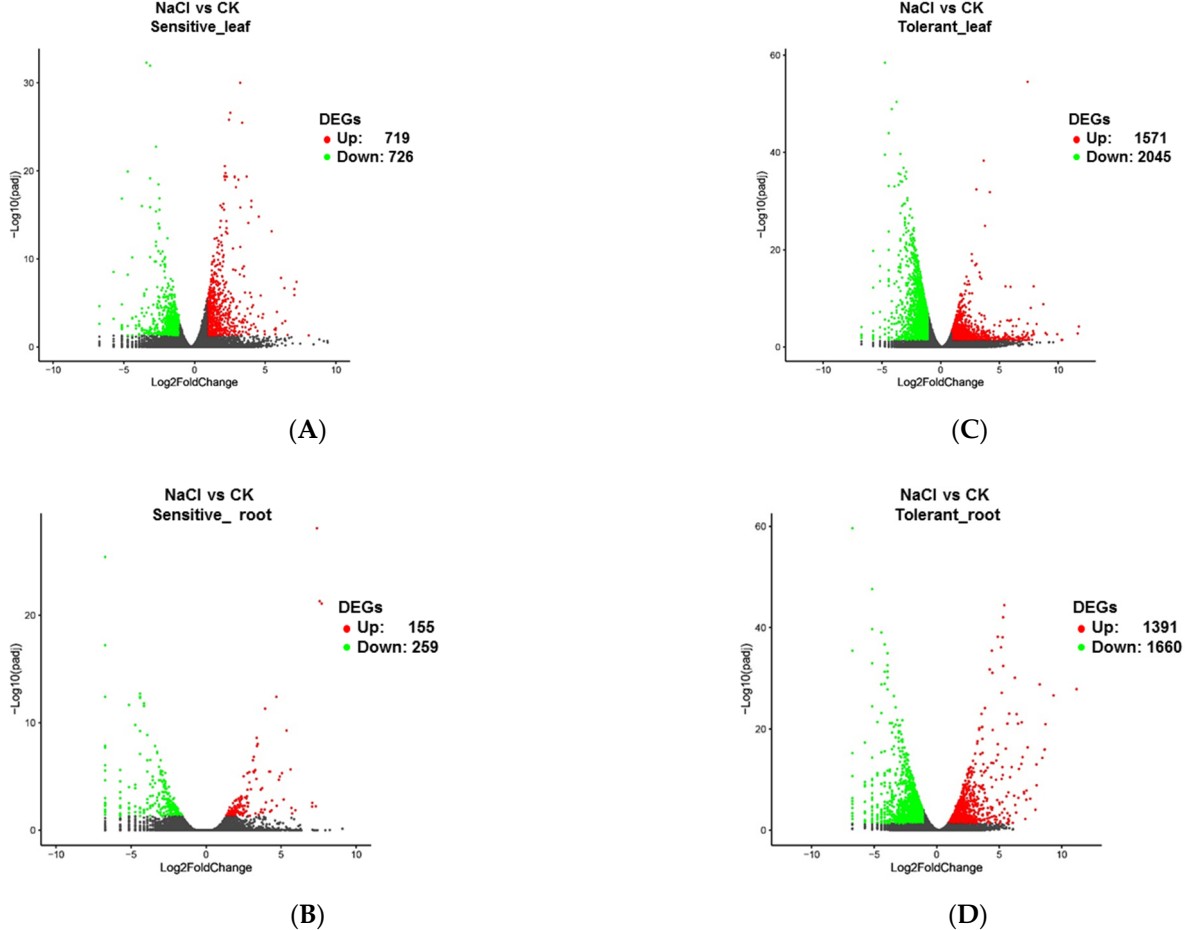

**Figure 3.** Volcano plots showing differentially expressed genes (DEGs) in leaf and root tissues in the salt-sensitive garlic accession (**A**,**B**) and salt-tolerant accession (**C**,**D**).The *x*-axis indicates log2 of fold change and the *y*-axis shows negative log10 of adjust p value. The plots were generated using an online platform (http://www.ehbio.com/Cloud_Platform/front/#/, accessed on 15 October 2020).

### 3.4. Functional Categorization of DEGs Identified in the Tolerant Leaves and Roots

A total of 5386 DEGs were then discovered in leaf and root tissues of the salt-tolerant accession by filtering the DEGs from the salt-sensitive accession, including 1282 genes upregulated only in leaves, 1068 upregulated only in roots, 113 upregulated in both leaves and roots, 1505 genes downregulated only in leaves, 1203 downregulated only in roots, and 215 downregulated in both leaves and roots (Figure 4).

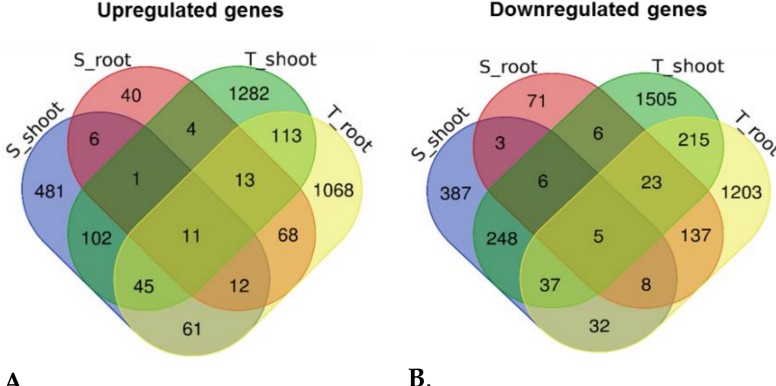

**Figure 4.** Common and different DEGs in the tolerant and sensitive leaves and roots. DEGs identified from the tolerant (**T**), and sensitive (**S**) leaves and roots were analyzed using online software (http://bioinformatics.psb.ugent.be/webtools/Venn/, accessed on 10 September 2020). Upregulated and downregulated genes by NaCl treatment were compared in (**A**,**B**), respectively.

Next, the DEGs were functionally categorized by GO (Gene Ontology) annotations. Fourteen biological processes were significantly upregulated in leaves. These processes included those associated with cellulose metabolism and microtubule dynamics, such as "cellulose metabolic process", "cellulose biosynthetic process", "beta-glucan metabolic process", "beta-glucan biosynthetic process", and "microtubule-based movement". In addition, the pathways involved in DNA metabolism and replication were enriched (Figure 5A and Supplementary Table S4). These upregulated pathways could enhance garlic salt tolerance since maintaining microtubules organization, cellulose synthesis, and DNA replication promoted biomass production or salt tolerance [47,48]. Sixty-one biological processes were upregulated in roots, such as "metabolic process", "oxidation-reduction process", and "transport" (Figure 5B and Supplementary Table S4). Meanwhile, 74 biological processes were downregulated in leaves, including "photosynthesis", "photosynthesis, light harvesting", "photosynthetic electron transport chain", "photosynthetic electron transport in photosystem I", "photosynthetic electron transport in photosystem II", and "photosystem II assembly" (Figure 5C and Supplementary Table S4), which was consistent with the observation that the photosynthesis rate was reduced in leaves under salinity. Only four biological processes were downregulated in roots, including "response to oxidative stress", "oxidation-reduction process", "response to stress", and "response to stimulus" (Figure 5D and Supplementary Table S4).

These genes were further analyzed using the Kyoto Encyclopedia of Genes and Genomes (KEGG) pathway enrichment to identify candidate genes controlling salt tolerance in garlic. The "ribosome biogenesis in eukaryotes", "DNA replication", "cell cycle", "phenylpropanoid biosynthesis", and "RNA transport" were major pathways upregulated solely in leaves. The "alpha-linolenic acid metabolism", "carotenoid biosynthesis", and "photosynthesis-antenna proteins" were significantly upregulated in roots (Figure 6A). The "galactose metabolism", "glutathione metabolism", "starch and sucrose metabolism", and "purine metabolism" were upregulated simultaneously in leaves and roots, although different genes in the same pathways were upregulated in these tissues (Supplementary Table S5).

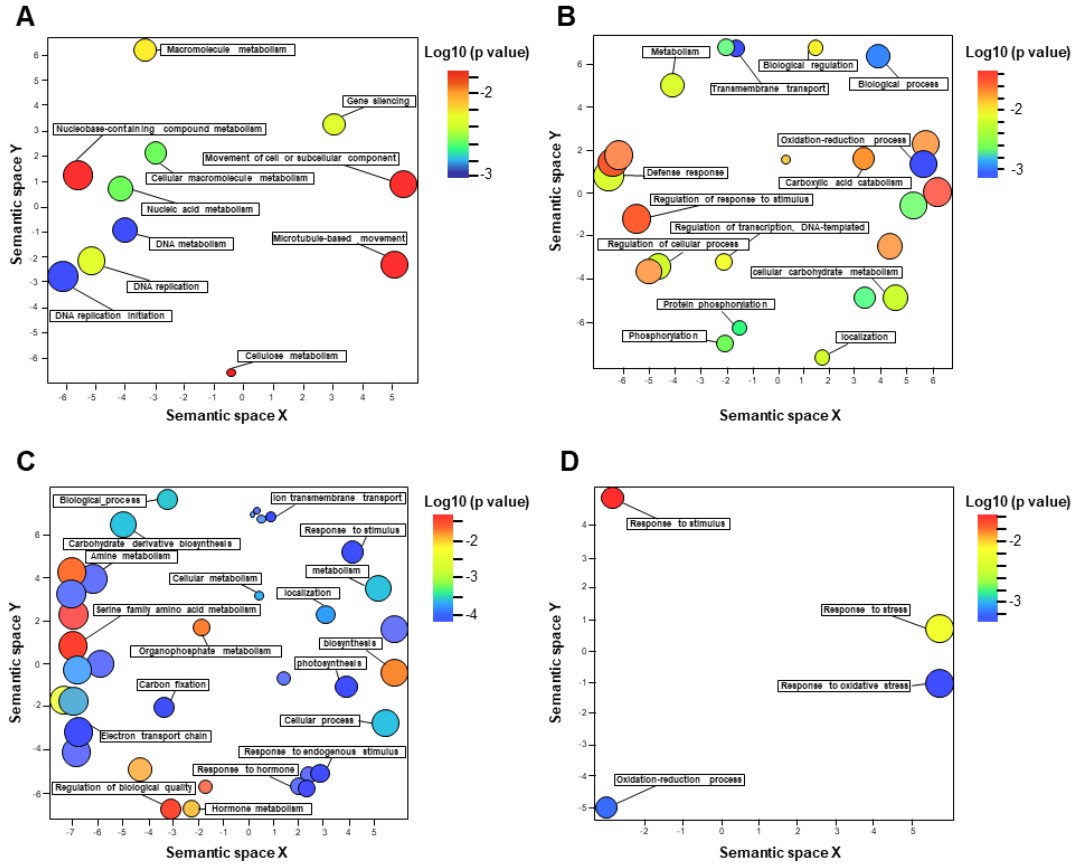

**Figure 5.** Functional categories of the DEGs identified in the salt−tolerant garlic accession. The upregulated biological processes in leaves (**A**) and roots (**B**), and downregulated biological processes in leaves (**C**) and roots (**D**) were visualized and clustered based on the semantic similarity using REVIGO. The circle color indicates the log10 of the adjusted *p*-value.

On the other hand, the "photosynthesis", "photosynthesis-antenna proteins", "carbon fixation in photosynthetic organisms", "porphyrin and chlorophyll metabolism", and "carbon metabolism" were downregulated solely in leaves, in line with the reduced photosynthesis rate in leaves under salinity (Figure 6B and Supplementary Table S5). The other downregulated pathways included "brassino-steroid biosynthesis", "glycine, serine and threonine metabolism", "isoquinoline alkaloid biosynthesis", and others. In comparison, the "plant hormone signal transduction", "plant MAPK signaling pathway", "nitrogen metabolism", "two-component system", and "betalain biosynthesis" were downregulated solely in roots. In addition, the "metabolic pathways" and "biosynthesis of secondary metabolites" were downregulated simultaneously in leaves and roots although different genes in the same pathways were regulated by NaCl treatment. Interestingly, several pathways were conversely regulated in leaves and roots, such as "photosynthesis-antenna proteins", "glutathione pathway", "phenylpropanoid biosynthesis", and "xenobiotics metabolism by cytochrome P450" (Supplemental Table S5).

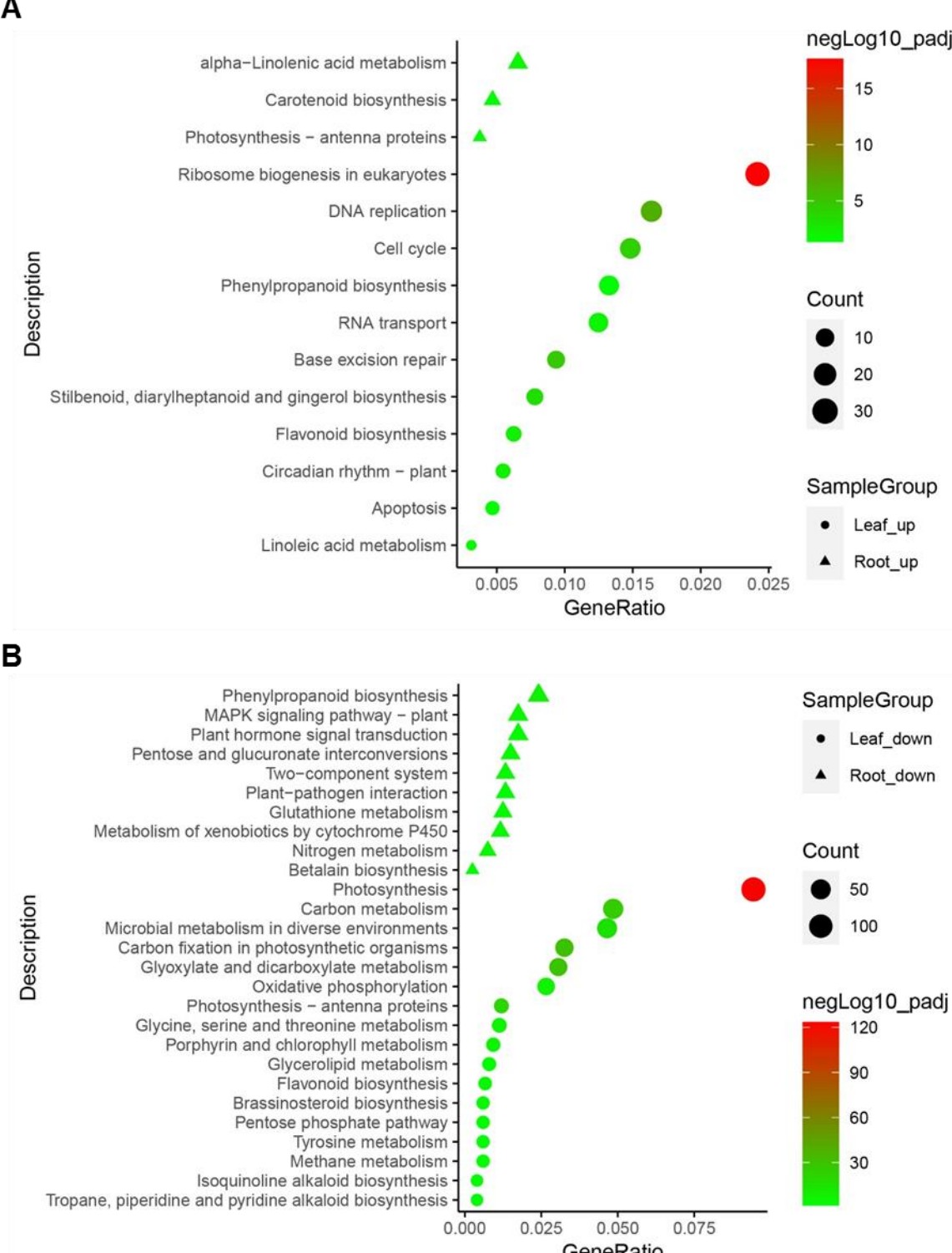

**Figure 6.** Exclusively regulated Kyoto Encyclopedia of Genes and Genomes (KEGG) pathways in leaves and roots. The circled area indicates the number of DEGs, and the circle color shows -log10 (adjusted *p*-values). The plots were generated using an online platform (http://www.ehbio.com/Cloud_Platform/front/#/, accessed on 15 October 2020). Upregulated and downregulated KEGG pathways were shown in (**A**,**B**), respectively.

### 3.5. Potassium Transporters were Significantly Upregulated in the Tolerant Roots

Reducing $Na^+$ concentration and increasing $K^+$ concentration in cells are essential mechanisms for plants to cope with salt stress. Plants have different $K^+$ transport systems that are responsive to $K^+$ uptake under various $K^+$ conditions [49]. Four genes (*Asa2G07164*, *Asa7G06244*, *Asa7G02557*, and *Asa8G02052*) encoding potassium transporters or potassium channel-like proteins were upregulated exclusively in the roots of the salt-tolerant cultivar. The similar expression patterns of these genes were confirmed by qRT-PCR though there were discrepancies between the fold changes, probably due to the limitations of the two methods (Table 2). In this case, the high $K^+/Na^+$ ratio in the salt-tolerant cultivar could be attributable to the elevated expression of potassium transporters in roots.

**Table 2.** Expression of possible potassium was upregulated in the tolerant roots.

| Gene ID | Annotation | Fold Change | Adjusted *p* Value | qRT-PCR |
|---|---|---|---|---|
| Asa2G07164 | Potassium channel like | 4.91 | 0.038099708 | 6.1 |
| Asa7G06244 | Potassium transporter | 2.72 | 0.008903446 | 3.7 |
| Asa7G02557 | Two-pore potassium channel like | 2.69 | 0.028014856 | 7.7 |
| Asa8G02052 | Potassium transporter | 2.36 | 0.039404635 | 4.4 |

### 3.6. Carotenoid Biosynthesis Pathway and Auxin Signaling Pathway Are Substantially Altered in the Tolerant Roots

Carotenoids are a large group of C40 pigments, some of which serve as precursors of phytohormones such as ABA and strigolactones [50]. Phytoene synthase (PSY), catalyzing the condensation of two geranylgeranyl diphosphate molecules to produce phytoene, is the key enzyme in carotenoid biosynthesis [51]. Lycopene beta-cyclase (LCYB) and beta-carotene hydroxylase (BCH) are essential enzymes in carotenoid synthesis [52–54]. In the tolerant roots, transcription of *PSY* (*Asa0G00778* & *Asa5G05878*), *LCYB* (*Asa5G05619*), and *BCH* (*Asa2G05330*) was significantly increased, indicating that the synthesis of carotenoids and ABA was enhanced in the tolerant roots after salt treatment (Figure 7A). However, *ABA 8′-hydroxylase* (*Asa1G00338*), which encodes an enzyme mediating ABA oxidation, was also upregulated. The increase of both ABA synthesis and oxidation suggests a high level of ABA metabolism in roots upon salt stress.

Auxin/Indole-3-Acetic Acid (Aux/IAA) proteins act as negative regulators of the early auxin response [55]. Small auxin-upregulated RNA (SAUR) proteins are a class of auxin-induced proteins [56], each of which could regulate different aspects of auxin-mediated growth and development [57]. In the tolerant roots, expression of *IAA3* (*Asa7G03113*) and *SAUR32* (*Asa7G00211* and *Asa8G05132*), *SAUR36* (*Asa0G03456*), and *SAUR71* (*Asa8G04231*) was reduced by salt treatment, increasing early auxin response and decreasing a subset of downstream auxin responses (Figure 7B).



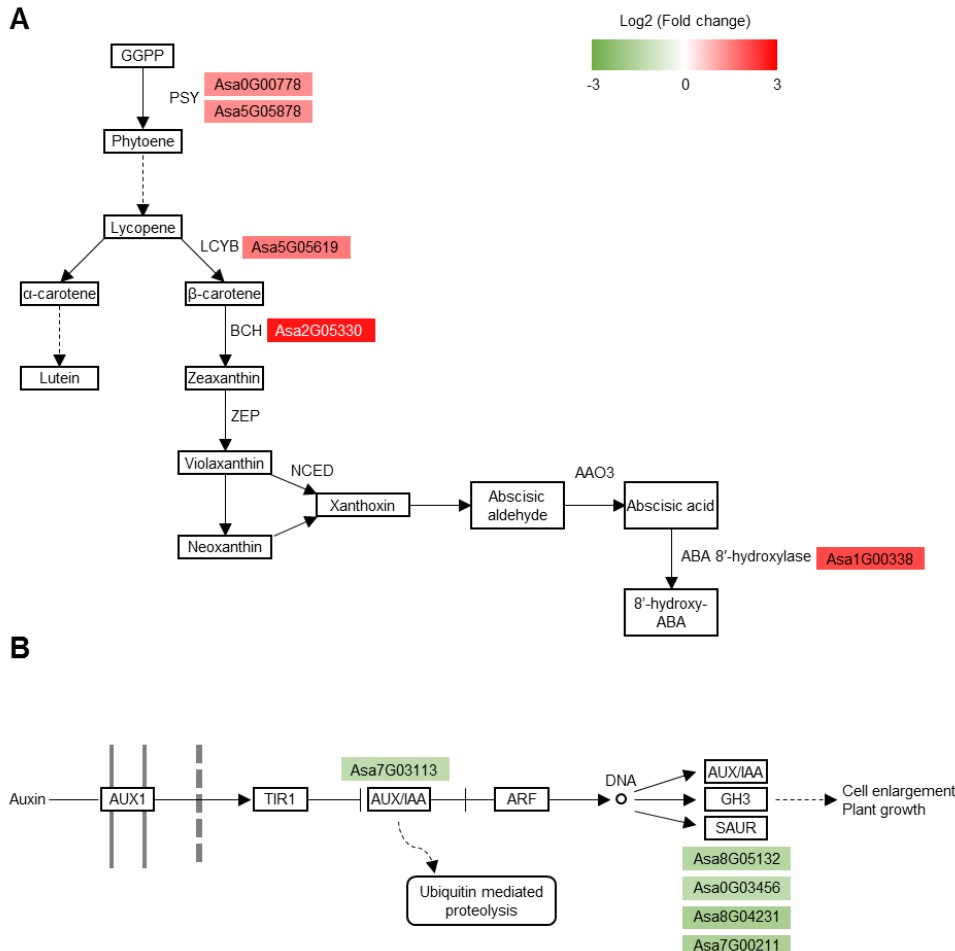

**Figure 7.** Altered expression of genes involved in carotenoid biosynthesis (**A**) and auxin signaling pathway (**B**). PSY, phytoene synthase; LCYB, lycopene beta−cyclase; BCH, beta−carotene hydroxylase; ZEP, zeaxanthin epoxidase; NCED, nine−cis−epoxycarotenoid dioxygenase; AAO3, aldehyde oxidase 3; AUX/IAA, Auxin/indole−3−acetic acid proteins; SAUR, small auxin−upregulated RNA proteins. The pathway maps were generated based on the KEGG database.

## 4. Discussion

This study assessed salt tolerance among a small group of soft-neck garlic accessions by morphological and physiological measurement and principal component analysis. The outcome demonstrated that PCA could be of great use in evaluating abiotic stress tolerance in a large population of garlic or other species. Furthermore, transcriptomic changes were examined in leaves and roots between a typical salt-tolerant and salt-sensitive accessions. This strategy remarkably reduced the number of DEGs reported from a previous study [32] and identified 5386 DEGs in salt-tolerant leaves and roots after NaCl treatment. Similar salt-regulated processes mediated by these DEGs were also found in earlier studies on garlic and other plant species, such as metabolic process, biosynthesis of secondary metabolites, starch and sucrose metabolism, purine metabolism, plant hormone signal transduction, plant MAPK signaling pathway, and others [32,58,59]. Meanwhile, this study revealed for the first time that root-specific carotenoid biosynthesis, auxin signaling, and $K^+$ transport were likely to be responsive to garlic salt tolerance.

$K^+$ and $Na^+$ transporters play an important role in salt tolerance by regulating the $K^+/Na^+$ ratio. Different $K^+$ transport systems are responsive to $K^+$ uptake under various environments [49]. For example, rice high-affinity $K^+$ transporter OsHAK5, is mainly expressed in root epidermis and stele, the vascular tissues, and mesophyll cell. Knockout of this gene reduced the $K^+/Na^+$ ratio in the leaves, resulting in hypersensitivity to salt

stress [60]. Four garlic genes (*Asa2G07164*, *Asa7G06244*, *Asa7G02557*, and *Asa8G02052*) encoding K$^+$ transporters or K$^+$ channel-like proteins were upregulated solely in the roots of the salt-tolerant cultivar, suggesting enhanced K$^+$ uptake in roots (Table 2). Thus, the upregulation of K$^+$ transporters expression in roots, consistent with the higher K$^+$/Na$^+$ ratio in the salt-tolerant garlic line, could be one of the mechanisms underlying garlic salt tolerance.

Salt stress caused upregulation of *PSY*, *BCH2*, and *zeaxanthin epoxidase* (*ZEP*) in Arabidopsis roots but not in shoots. As a result, ABA levels in stressed Arabidopsis roots and plant salt tolerance were enhanced, demonstrating that elevated expression of carotenoid biosynthetic genes is crucial to ABA biosynthesis in roots and salt tolerance [16]. Garlic *PSY*, *LCYB*, and *BCH* were upregulated exclusively in roots by salt exposure, indicating a similar mechanism used by this plant to cope with salinity stress. The basic expression level of *PSY* and *BCH* was much higher in garlic leaves than in roots (Supplementary Table S3); it is hard to detect alterations in their expression by salt treatment in the entire plant. Separation of root- and shoot-specific DEGs resulted in identifying root-specific salt tolerance mechanisms in garlic.

AUX/IAA proteins serve as negative regulators of early auxin response by binding auxin response factors (ARF) to prevent the expression of auxin-responsive genes in the absence of auxin [61]. A high level of auxin causes degradation of AUX/IAA proteins via the 26S proteasome and releases ARFs to activate auxin-responsive genes [62]. In garlic tolerant roots, *IAA3* (*Asa7G03113*) was downregulated by salt, suggesting enhanced early auxin response.

More than 70 SAUR proteins were identified from the Arabidopsis genome. Several SAUR proteins were involved in development and growth, such as apical hook development, hypocotyl elongation, and later root development [57,63–65]. In wheat, the *TaSAUR75* gene was downregulated by salt in roots, and overexpression of *TaSAUR75* led to higher root length and survival rate under salt and drought stress, suggesting involvement of SAUR proteins in response to abiotic stresses [66]. Garlic *SAUR* genes (*Asa7G00211*, *Asa8G05132*, *Asa0G03456*, and *Asa8G04231*) were downregulated by salt treatment in roots. Although little information is available on these genes' physiological function, the converse regulation of transcription of the auxin signaling components and the enhanced ABA biosynthesis could finely tune the appropriate development and physiology of garlic roots under salt stress. Further analysis of their functions and involvement in diverse metabolic pathways would deepen our understanding of garlic-specific and tissue-specific mechanisms underlying salt tolerance.

## 5. Conclusions

Examination of garlic responses and gene expression in leaves and roots under salt stress by combining principal component analysis and RNA-seq revealed an essential role of root-specific carotenoid biosynthesis, auxin signaling, and K$^+$ transport in garlic salt tolerance.

**Supplementary Materials:** The following are available online at https://www.mdpi.com/article/10.3390/agronomy11040691/s1, Table S1: Primers used in the study, Table S2: Summary statistics of the RNA-seq results, Table S3: DEGs identified in the salt-tolerant accession, Table S4: GO terms analysis of the DEGs from the salt-tolerant accession, Table S5: KEGG enrichment analysis of the DEGs from the salt-tolerant accession.

**Author Contributions:** X.Z. and C.Z. designed the research. X.Z., Y.D., X.H., G.L., H.Z., D.J., J.F., J.A.C.-A., X.L., and S.J.C.-P. performed the experiments. R.M.B., M.E.M.A., H.O.L., H.A.D.S., and M.V.V. provided technical assistance. X.Z. and C.Z. wrote the manuscript. All authors have read and agreed to the published version of the manuscript.

**Funding:** This work was supported by Nexus Crops CRISPR and Phenomics: Individual and Institutional Capacity Building for Crop Science at Universidad Nacional de San Agustin (UNSA), Peru and the USDA National Institute of Food and Agriculture (Hatch project 1007567).

**Data Availability Statement:** RNA-Seq data in this study have been deposited in NCBI with accession no. PRJNA682570.

**Acknowledgments:** We are grateful to the Regional Plant Introduction Station, WRPIS, USDA-ARS, for providing garlic accessions. We thank Zhangjun Fei and Michael Lu for their help with bioinformatics and principal component analysis.

**Conflicts of Interest:** The authors declare no conflict of interest.

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
