# Peer review of "Using Principal Component Analysis and RNA-Seq to Identify Candidate Genes That Control Salt Tolerance in Garlic (Allium sativum L.)"

_agronomy, doi:10.3390/agronomy11040691_

Round 1
Reviewer 1 Report
In this study, the author uses RNA-Seq to Identify Candidate Genes that Control Salt Tolerance in Garlic (Allium sativum L.). From 17-day-old seedlings of eight accessions treated with 200 mM NaCl for seven days in a hydroponic system. Several morphological and physiological traits were measured at the end of the treatment, including shoot height, shoot fresh weight, shoot dry weight, root length, fresh root weight, root dry weight, photosynthesis rate, and concentrations of Na+ and K+ in leaves. The principal component analysis showed that dry shoot weight and K+/Na+ ratio contribute the most to salt tolerance among the garlic accessions. As a result, salt-tolerant and sensitive accessions were grouped based on these two parameters. To investigate the molecular mechanisms in garlic in response to salinity, the transcriptomes of leaves and roots between salt-tolerant and salt-sensitive garlic accessions were compared. 1.5 billion read pairs were obtained from 24 libraries generated from the leaves and roots of the salt-tolerant and salt-sensitive garlic accessions. A total of 47,509 genes were identified by mapping the cleaned reads to the garlic reference genome. Statistical analysis indicated that 1,282 and 1,068 genes were upregulated solely in the tolerant leaves and roots, whereas 1,505 and 1,203 genes were downregulated exclusively in the tolerant leaves and roots after NaCl treatment, respectively. The manuscript is very well written. However, for the manuscript's betterment, I would like to make few suggestions to the authors.
- The authors need to validated RNA seq data by qRT-PCR analysis—especially the genes in Table 2.
- The author should functionally analyze at least one gene from table two for its possible salt stress tolerance role.
- The position of figure 1 is not correct. It should not be in the materials and methods section; rather, it should be in the results section.
- Change at
L58 ROS full form?
L107 were taken at seven days to were taken seven days.
L110 to final concentration to to a final concentration.
L137 mismatsches to mismatches.
L143 DEGs full form?
L286 circle area to circled area.
L344 salt tolerance were to salt tolerance was.
I found plagiarism at these lines L21, L117, L140, L153-154, L228-230, L250-251, L256-258, L262, L294-295, L339-340. Please clean it.
Reviewer 2 Report
The manuscript includit evaluation of garlic genotype and phenotype in response to salt stress. It is a very interesting study, but not without a few flaws and inaccuracies.They are as following:
- In the introduction, there is no justification why it is important for this species to search for salt tolerant genotypes. Whether the soil in an important area for the cultivation of this species is highly saline.
- The principal componets analysis is standrad methods used to the evalauation of phenotypic traits. Doing from “using Principal Component Analysis” or “Application of Principal Component Analysis” the title or aime of the manuscript is nothing new. I would suggest changing the title and aim.
- It does not clearly follow from the content of the Material and methods chapter that the PCA method was used for phenotypic traits.
- Line 91-92. How many accesions were assessed in the study in total? Unfortunately, as a reviewer, I had limited access to additional materials. I couldn't unpack them properly in Windows. Such information would come in handy in the Material and Methods chapter.
- There is no precise information during which experiment the phenotypic traits were assessed, whether it was a field experiment or just a laboratory one. And in what experimental setup.
- Line 124. The link goes to a description of the use of Principal Component Analysis in an R program, and certainly not a R package. Besides, this is a page in Chinese that has not been understood by the rest of the scientific world. The proposed links should be more universal, especially for software.
- Line 125. The PCA method certainly does not allow garlic genotypes to be grouped. Just to judge their similarities. There are other statistical methods for grouping objects (eg Cluster analysis).
- In Table 1 full names of the traits should be provided (I suspect that these are phenotypic)
- Line 174-176. The numbers of the objects do not tell the reader anything (they are in the supplement), in this case it is worth giving their full names and origin.
- Figure 2A. The reader will certainly not be interested in the detailed results of the PCA analysis, especially the proprietary values for all the main indexes, especially since the authors focus on the first two components anyway.
- Figure 2C. He suspects this is a classic scatterplot for PCA. But it doesn't look much like, where the 0 values for the first and second components are. There is also no explanation for the division for tolerance and sensitive genotypes.
- It is strange that with the information about the phenotype and genotype, the authors did not try to analyze the QTL type.
Round 2
Reviewer 1 Report
I am happy with the comments and the Manuscript can be accepted in its current format.
Reviewer 2 Report
All comments have been included in the new version of the manuscript